# Parents' Evaluation of Developmental Status and Strength and Difficulties Questionnaire as Screening Measures for Children in India: A Scoping Review

**Hina Sheel** , **Lidia Suárez** and **Nigel V. Marsh** *

School of Social and Health Sciences, James Cook University, 149 Sims Drive, Singapore 387380, Singapore
* Correspondence: nigel.marsh@jcu.edu.au

**Abstract:** Due to the limited availability of suitable measures, screening children for developmental delays and social–emotional learning has long been a challenge in India. This scoping review examined the use of the Parents' Evaluation of Developmental Status (PEDS), PEDS: Developmental Milestones (PEDS:DM), and the Strength and Difficulties Questionnaire (SDQ) with children (<13 years old) in India. The scoping review was conducted following the Joanna Briggs Institute Protocol to identify primary research studies that examined the use of the PEDS, PEDS:DM, and SDQ in India between 1990 and 2020. A total of seven studies for the PEDS and eight studies for the SDQ were identified for inclusion in the review. There were no studies using the PEDS:DM. Two empirical studies used the PEDS, while seven empirical studies used the SDQ. This review represents the first step in understanding the use of screening tools with children in India.

**Keywords:** PEDS; SDQ; screening; children; India; scoping review

## 1. Introduction

Developmental disability (DD) is a broad spectrum of impairments or a lack of developmental features appropriate to a child's age and vital for their growth [1]. DD is usually present at birth and negatively impacts the individual's physical, intellectual, and/or social development. DD results in impairments in the ability to learn, reason, and solve problems, and it also impairs adaptive behaviour, which consists of social and life skills [2]. The *Diagnostic and Statistical Manual of Mental Disorders, 5th edition, text revision* (DSM-5-TR) characterised DD as an Intellectual Developmental Disorder (IDD), where an individual lacks in general mental abilities and adaptive functioning [3]. The *International Classification of Diseases, Eleventh Revision* (ICD-11) considered DD as a neurodevelopmental disorder that arises during the developmental period and involves difficulties in intellectual, motor, language, or social functions [4].

Social–emotional learning (SEL) is a child's ability to understand themselves and others, regulate emotions and attention, and engage with others [5]. Individuals with DD may have difficulties with social relationships when compared to neurotypical peers and have differences in their reading of neurotypical nonverbal and subtle social cues [6]. Therefore, they usually have impairments in SEL too.

In India, 2.5–3.4% of children had various developmental problems diagnosed using screening tools. The most common forms were developmental delay, speech delay, global delay, gross motor delay, and hearing impairment [7]. In addition, in India, a review of recent studies showed that the prevalence of mental health problems in school-going children varies from 6.33% to 43.1%. Specifically, the prevalence of behavioural and emotional problems among orphans and other vulnerable children ranges from 18.3% to 64.53%. In children with typical development, it was reported to range between 8.7% and 18.7% [8].

The recent research literature in India has revealed multiple issues within universal developmental and SEL surveillance and screening [9,10]. To begin with, parents are unaware that screening services exist, nor are they aware of why those services are necessary. Health care is given priority only when there is an acute illness. Furthermore, the population of doctors who serve the needs of Indian children is heterogeneous, with varying skills. If parents express concerns, they often receive inaccurate information without proper evaluation [10]. Postgraduate paediatric courses in India lack formal training in developmental and SEL screening and assessment [10]. Moreover, in India, paediatricians make clinical judgments based on unstructured probing of developmental milestones, and India needs more developmental paediatricians [10,11].

In 2013, the Indian government launched the Rashtriya Bal Swasthya Karyakram, also known as the 'Child Health Screening and Early Intervention Services scheme', which caters specifically to government schools [12]. The scheme aims at early identification and early intervention for children from birth to 18 years old to cover the four Ds: defects at birth; deficiencies; diseases; and development delays. This includes disability. The target population included new-borns, children in Anganwadi centres (rural childcare centres across India), and government schools [13]. However, the annual progress report of the scheme for 2018–2019 provided scant information on the tools used for screening purposes. Mukherjee et al. [11] concluded that the number of children identified for delay and disability has increased since the inception of the scheme. However, some states, such as Maharashtra and Odisha, faced issues with implementation, infrastructure constraints, and limited resources [14]. Furthermore, the scheme does not cater to private schools in India, which constitute 49% of children in urban areas and 21% in rural areas [15].

Developmental screening "is a brief assessment procedure designed to identify children who should receive a more intensive diagnosis or assessment" [16]. Developmental surveillance monitors the child's progress by gathering information on the child's development from multiple sources and determining whether the rate and extent of a child's development elicits concerns [17].

The World Health Organization (WHO) has emphasised the importance of screening children for any form of disability, explicitly highlighting the relevance of interventions that promote young children's development [18]. Screening tools developed in India, such as the Baroda Development Screening Test (BDST), Developmental Assessment Scale for Indian Infants (DASII), and Trivandrum Developmental Screening Chart (TDSC), are linguistically and culturally reasonable. Nonetheless, their psychometric properties are suboptimal, and their use has been restricted to a specific population given that health professionals initially developed them for community services [10,19].

Most low- and medium-income countries (LMIC) use tools developed in Western countries to screen children for DD and SEL. However, there are three main limitations to using these screening tools. First, most screening tools measuring DD in children aged 0–8 years are developed in Western and high-income countries. They require extensive training, which is not readily available in LMIC due to limited funds to purchase tools and training costs [20]. Second, tools developed in Western countries lack psychometrically valid translations to use in other cultures [21,22]. Third, most screening tools are copyrighted and require permission to translate them into other languages for schools and clinics to use; this is often expensive [23].

Screening tools that have been adapted and translated for use in LMIC include the Bayley Scale of Infant and Toddler Developmental Screening test (BSITDS; ref. [24]), Ages and Stages Questionnaire (ASQ; ref. [25]), Guide for Monitoring Child Development (GMCD; ref. [26]), and Parent Evaluation of Developmental Status (PEDS; ref. [27]) for assessing DD. The Eyberg Child Behaviour Inventory (ECBI, ref. [28]), Child Behaviour Checklist (CBC; ref. [29]), Children Emotional Adjustment Scale (CEAS; ref. [30]), and Strength and Difficulties Questionnaire (SDQ; ref. [31]) have been adapted and translated for assessing SEL.

In comparison to the PEDS, PEDS:DM, and SDQ, the other screening tools reviewed such as the BSITDS, Denver Developmental Screening Test (DDST), ASQ-3, GMCD, ECBI, and CEAS have several limitations [22,24–26,28,30,32]. First, these scales' psychometric properties have been questioned for LMIC because research outcomes from the Western world cannot be applied to LMIC [33,34]. Second, tools such as the BSITDS and GMCD require professional training, which is time consuming and costly [35]. Finally, scales such as the ASQ-3 and BSITDS-III need parents and clinicians to attempt multiple developmental tasks with the child before filling in the questionnaire, which may hinder the evaluation due to the longer administration time and the child's level of comfort with the activity and the environment [36,37].

Among the screening tools for DD and SEL, the PEDS and SDQ are probably the most appropriate for use in India because they are less costly and cater to a wider age range (compared to tools such as GMCD and CEAS). The PEDS and SDQ are also easily accessible, do not require extensive training for administration, and have proven psychometric properties for use in LMIC [31,38]. Some preliminary studies have validated PEDS in LMIC such as Thailand [39], Bhutan [40], Tehran [41], and India [42,43], and the SDQ has been validated in Nigeria [44], Vietnam [45], Turkey [46], Thailand [47], and India [48–50]. However, the PEDS studies carried out in India were challenged by the PEDS developer for its scoring procedure and the gold standard tool the study used for its cross-validation [42,43,51]. Furthermore, these studies did not use Parents' Evaluation of Developmental Status: Developmental Milestone (PEDS:DM). Philips Owen et al. [49] translated and validated the SDQ in the regional language (Malayalam) instead of the Indian national language of Hindi. Michelson et al. [48] and Singh et al. [50] used the Hindi version of the SDQ in their respective studies. Nevertheless, these studies validated the Hindi SDQ on adolescents and did not consider children. Therefore, there is limited research on whether these measures have been translated, adapted, and validated with Indian children [21,47].

The PEDS [27] is a surveillance and screening tool for children aged 0 to 8 years. The tool elicits and addresses parents' concerns about development, behaviour, and mental health. The tool comprises one form with 10 questions across 10 categories (expressive language, receptive language, social–emotional, behavioural, fine motor, gross motor, self-help, school, cognitive, and health). The questions in the PEDS elicit parents' perspectives of their child's development as high/medium/low risk. The response options include yes, no, and a little. The scoring for the PEDS includes columns for each age range and identifies which concerns predict problems and which do not. The PEDS interpretation form houses an algorithm to decide whether to refer, screen further, observe, counsel parents, or reassure them on the results obtained [27]. PEDS has sound psychometric properties and was re-standardised and revalidated in 2013 [52]. The interrater reliability was 0.95, and the test–retest reliability was 0.88. The validity of the PEDS ranges from 0.84–0.99 when compared with later deficits and diagnoses [52].

The PEDS:DM is a new measure that can be used with the PEDS or by itself. The PEDS:DM comprises six to eight items per age and aims to predict the developmental status of children accurately. Each item on the PEDS:DM addresses different domains (fine motor, gross motor, expressive language, receptive language, self-help, social–emotional, and, for older children, reading and math). The age-appropriate items are presented on a single page within a laminated book that includes essential visual stimuli. Parents answer the PEDS:DM items via a multiple-choice format in fewer than 5 min. A single scoring template that is built into the binder is used to determine whether the milestones are met or unmet. Furthermore, the PEDS:DM uses the same evidence-based decision regarding the results as the PEDS. The PEDS:DM is reported to have good psychometric properties, with internal consistency across all domains being 0.98. The test–retest reliability was 0.98 and 0.99, and the interrater reliability is reported to range from 0.82 to 0.96 across subtests. The concurrent, discriminant, and criterion-related validity for PEDS:DM is satisfactory compared with other similar disability and screening tools. In addition, the specificity and sensitivity of the scale are 80% and 85%, respectively, indicating that PEDS:DM reports few

false negative results. Thus, fewer children with developmental disabilities were missed in addition to correctly identifying children with no delays or disabilities [53].

The SDQ screening measure evaluates children's mental health problems in the age range of 2–16 years [54]. The SDQ is completed by parents and teachers and comprises 25 questions under five domains: emotional symptoms, conduct problems, hyperactivity/inattention, peer relation problems, and prosocial behaviour [31]. This screening tool comprises a three-point rating scale ranging from not true, somewhat true, and certainly true. The scoring for the SDQ comprises the total difficulties score, which is obtained by summing the scores for all scales except the prosocial scale. The resulting scores range from 0 to 40. The cut-off points for SDQ scores are categorised as normal, borderline, and abnormal [54]. The SDQ has good psychometric properties. The tool was administered to 10,435 British participants by their parents, teachers, and self-evaluation. The internal consistency of the tool was 0.73, the test–retest reliability was 0.62, and the sensitivity and specificity of the scale were 95% and 35%, respectively [54]. The SDQ reported high sensitivity, i.e., the tool correctly identified individuals 95% of the time with mental health problems. However, the SDQ inaccurately identified participants with no mental health problems as false positives, as reported with low specificity. There have been recent attempts to examine the usefulness of the SDQ for children with DD [55].

## 2. Objective and Research Question for the Scoping Review

Scoping reviews are used in healthcare research to map the scope and depth of a concept in a specific research area and to identify the sources and types of evidence available [56]. This scoping review's primary objective is to determine the extent to which two developmental screening tools (PEDS and PEDS:DM) and one SEL screening tool (SDQ) have been used with children in India. This review aims to do the following: first, increase awareness among parents and professionals in health and education about the relevance of screening children for DD and SEL. Incorporating screening tools during doctor visits and school enrolments may result in earlier and more rigorous assessment and intervention. Second, it aims to promote the use of valid, reliable, and accessible low-cost tools in LMIC such as India. Third, since PEDS, PEDS:DM, and SDQ meet the criteria, the study aims to determine whether these tools are validated for use in India.

The scoping review addresses the following three research questions: (1) what is the published evidence for the use of the PEDS, PEDS:DM, and SDQ in screening children aged 0–12 years for DD and SEL in India, (2) what are the demographic characteristics of the studies' participants, and (3) what conclusions have been drawn from the empirical research using the PEDS and SDQ screening tools in India?

## 3. Inclusion and Exclusion Criteria

This scoping review was completed using the PRISMA extension for scoping reviews and the Joanna Briggs Institute (JBI) Protocol for evidence synthesis (Appendix B) [57].

### 3.1. Population

The study included children aged 0–12 years and living in India. Exclusion criteria included studies conducted on people older than 12 years.

### 3.2. Concepts

Studies included in this review had to use the PEDS, PEDS:DM, or SDQ as screening tools. Only studies written and published in English and between the years 1990 and 2020 were considered for this review.

### 3.3. Context

The context of this review was limited to studies conducted in India. However, the setting across India could vary from children in schools to orphanages or institutional homes.

### *3.4. Types of Sources*

Primary research studies, systematic reviews, meta-analyses, experimental studies and epidemiological (grey literature) research were included in this scoping review.

## 4. Search Strategy

### *4.1. Pre-Identification Process*

The pre-identification process consisted of identifying and refining the research question. In the current scoping review, three questions were developed to explore whether the PEDS and SDQ screening tools have been used with the population of India, the participants' demographic characteristics, and the findings obtained from these studies.

### *4.2. Identification Stage*

The identification stage involved identifying relevant studies published between 1990 and 2020 using databases such as the Web of Science, Scopus, PEDS and SDQ websites, and Google Scholar for grey literature. Different variations of keywords were included. For the PEDS, the keywords were Parent Evaluation of Developmental Status; PEDS; Parent Evaluation of Developmental Status: Developmental Milestones; PEDS:DM; children; and India. For the SDQ, the keywords included Strength and Difficulties Questionnaire; SDQ; children; and India. The grey literature was also searched using Google Scholar. The literature included consensus, opinion, and position papers. A total of 61 articles for the PEDS (PEDS website = 28, Web of Science = 3, Scopus = 8, and Google Scholar = 22) and 184 articles for the SDQ (SDQ website = 13, Web of Science = 22, Scopus = 29, and Google Scholar = 120) were identified for possible inclusion in the scoping review. The identification stage processes were conducted by the first author (HS).

### *4.3. Screening Stage*

The first author (HS) carried out the initial screening and intentionally maintained the screening process as inclusive. In this stage, Endnote was used to accumulate all articles (61 articles for the PEDS and 184 articles for the SDQ) identified through the different databases and removed duplicate articles. The PEDS:DM is a new measure recommended to be used with the PEDS. However, the PEDS:DM was not mentioned in any articles in the initial screening. Therefore, it was eliminated from the PRISMA diagram. Relevant titles and articles that mentioned DD and SEL were retained. The first and third authors (HS and NVM) then conducted 100% of the screening using the inclusion and exclusion criteria. The JBI System for the Unified Management, Assessment, and Review of Information (JBI SUMARI) was used to gather and screen all the articles for the PEDS and SDQ.

### *4.4. Eligibility Stage*

The second screening stage involved screening titles and abstracts to determine the use of the PEDS and SDQ in India. At this stage, 38 articles from the PEDS collection and 132 articles from the SDQ collection were removed for not meeting the inclusion criteria.

The first and third authors conducted 100% of the screening. Overall, an interrater reliability of 98% was obtained between the two reviewers on the agreement of including 20 full-text articles for the PEDS and 29 articles for the SDQ for review. The 2% disagreement between the reviewers was resolved by reviewing and discussing the articles again. The reviewers agreed to not include these articles in the scoping review.

### *4.5. Final Screening Stage*

At this stage, only studies such as empirical research, systematic reviews, literature reviews, and grey literature (dissertation, opinion pieces, etc.) that mentioned the use of the PEDS and SDQ in India were incorporated into the scoping review. For the review decision process, the first author (HS) reviewed all the full-text articles.

The reviewer excluded full-text articles that: (1) did not use the PEDS or SDQ in India, (2) used the PEDS or SDQ on adolescents, or (3) did not provide sufficient information on

if, and if so how, the tools were translated to Hindi. As a result, 13 full-text articles were removed from the PEDS collection, and 21 articles were removed from the SDQ collection. A total of seven articles (six peer reviewed and one grey literature) were included in the final review for the PEDS, and eight articles (six peer reviewed and two grey literature) were included in the final review for the SDQ (Figures 1 and 2).

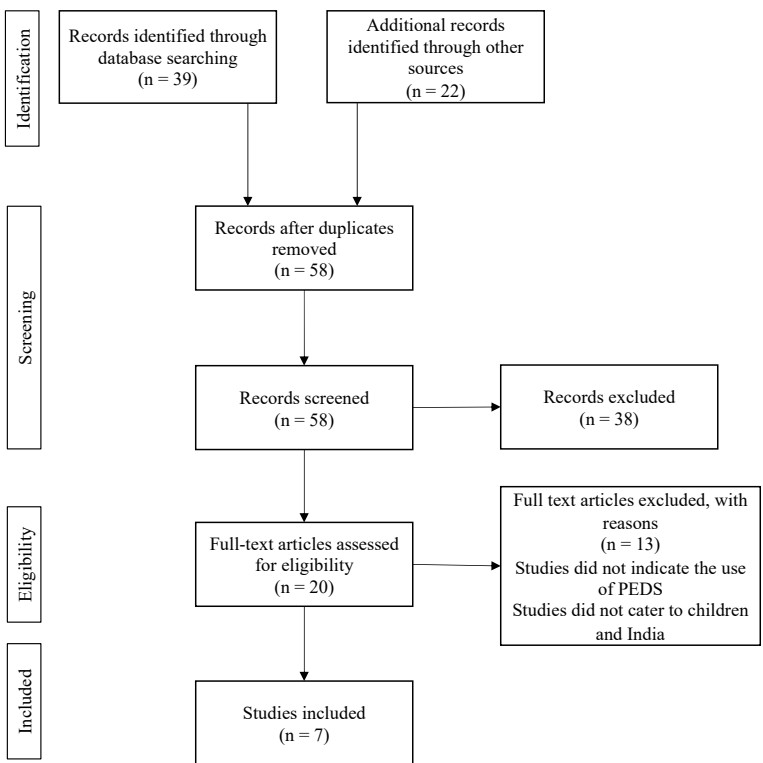

**Figure 1.** PRISMA diagram of studies included in the comprehensive scoping review for the PEDS.

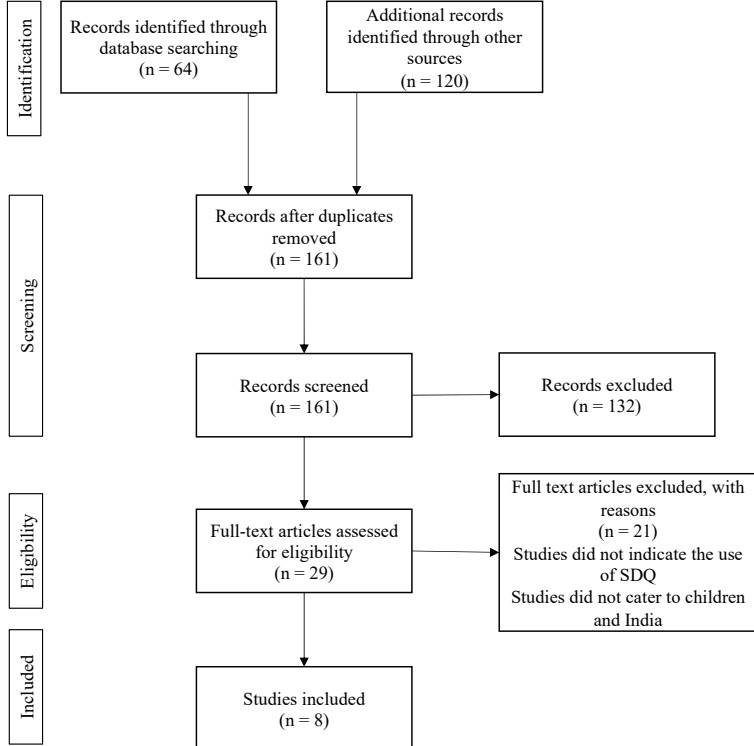

**Figure 2.** PRISMA diagram of studies included in the comprehensive scoping review for the SDQ.

## 5. Data Extraction Process

The following information was extracted from the seven articles for the PEDS and the eight articles for the SDQ: (1) author and date of publication, (2) type of article, (3) type of study, (4) where the study was conducted in India, (5) aim of the study, (6) the sample size and age range, (7) the setting in which the PEDS and SDQ were used, and (8) the main findings of the study (Appendix A).

## 6. Results and Discussion of the Scoping Review

The aim of this review was to examine the amount of published evidence on the use of the PEDS, PEDS:DM, and SDQ for screening children for DD and SEL in India, to explore the demographic characteristics of the studies' participants, and to report the conclusions drawn from the empirical research using the PEDS and SDQ screening tools.

## 7. PEDS and PEDS:DM

The scoping review for the PEDS found seven full-text articles on the use of the PEDS for DD in India. However, only two studies were empirical research, and none used the PEDS:DM, either alone or with the PEDS. The participants belonged to north India and were between 6 and 60 months of age. The conclusion drawn was that the PEDS could detect concerns among parents regarding their child's developmental milestones.

The number of PEDS studies varied across years. No studies were reported for 1990–2000, three (43%) were reported for 2001–2005, one (14%) was reported for 2006–2010, two (29%) were reported for 2011–2015, and one (14%) was reported for 2016–2020. Three (42%) of the studies were text- and opinion-based evidence, two (29%) were empirical studies, and two (29%) were systematic reviews.

Malhi and Singhi [42,43] reported the results from two empirical studies using the PEDS. These two studies did not use the PEDS:DM and aimed to identify the range of concerns that parents have about their children's development and its relationship to the child's developmental status. The first study included 55 parent–child dyads recruited through outpatient paediatric care in a tertiary care teaching hospital in Chandigarh. The second study recruited 79 parent–child dyads from the same hospital and city in India. The age range of the children was 6–60 months. The first study concluded that 38% of the parents indicated no concerns, while 20% raised non-significant developmental problems. Among these children, 91% passed the developmental screening. The second study concluded that parents' concerns regarding their children's developmental milestones were moderately sensitive predictors of DD in children aged 2 to 5 years. The authors suggested that since the PEDS's specificity (34.8%) and sensitivity (65%) were lower in LMIC than in the United States, the tool is not recommended as an alternative to standardised screening measures. Instead, it may be used as a pre-screening tool in an outpatient setting to identify those children who require more in-depth developmental screening [43].

In response to the conclusions of Malhi and Singhi [41,42], Glascoe [51] wrote a letter to the editors indicating that their findings may not be accurate for two reasons. First, there was a lack of clarity regarding their scoring of the PEDS. Second, their use of the Developmental Profile-II as the criterion measure was problematic, as it tends to both under- and over-detect developmental concerns among children [51].

Poon et al. [58] provided an opinion piece discussing the prevalence of DD in children. This article emphasised the benefits of early identification using developmental screening and surveillance. The authors believed that it is necessary to listen to parents' concerns with regularity, integrate routine screening with health maintenance visits, refer patients to paediatricians and therapists early, and provide early intervention services and therapies that have proven effective independent of the medical diagnosis. The authors reported that the PEDS has been translated into more than 10 different languages and completed by parents. Mukherjee et al. [10] also found the PEDS to be reliable in developing countries. However, the authors concluded that there is limited research from India.

Marlow et al. [37] conducted a systematic review of DD and autism spectrum disorder (ASD) screening tools and provided DD and ASD screening recommendations for LMIC. The review included children aged 0–7 years, studies published in English, the tools used for screening purposes, studies that included at least one of the developmental domains, and provided information on the measure's performance. The review confirmed that the PEDS has been translated for use in India and that it can detect DD among children in LMIC.

Woolfenden et al. [59] aimed to understand the use of the PEDS in evaluating parental concerns of children with developmental risk and associated risk factors. Their systematic review's inclusion criteria specified primary observational studies with available prevalence data. Their review found that the PEDS reported 13% of parents indicating their child as high developmental risk and 19% of parents reporting moderate to low developmental risks. However, these evaluations depended on the children's body weights, socioeconomic conditions, and access to medical care, which provided variation in the quality of studies included in the systematic review. Furthermore, comparing the PEDS to other measures of developmental risk such as the DDST, ASQ, and the Australian Early Developmental Index showed the same confidence interval around the pooled prevalence estimates of high and moderate developmental risks. Therefore, the two systematic reviews concluded that there is a substantial literature on the prevalence of parental concern for developmental risk in children. However, most of the findings were flawed due to the methodological issues of a small sample size and the use of an inappropriate measure to screen children for developmental delay. Furthermore, only eight studies used the PEDS in LMIC, including India. Table 1 summarises the studies conducted using PEDS in India. The table reports the authors, the article type, the design of the study, the city/state where the study was conducted, the aim of the study, the population setting (size and age range), the measures used, the results, and the key findings of the study.

**Table 1.** Summary of studies on Parent Evaluation of Developmental Status (PEDS) use in India.

| Author | Article Type | Design | City in India | Aim/Purpose | Population | | Setting and Measures Used | Main Results | Key Findings |
|---|---|---|---|---|---|---|---|---|---|
| | | | | | Size | Age Range | | | |
| Marlow, Servili, and Tomlinson, 2019 [37] | Peer reviewed | Systematic review | Chandigarh, India (Reference to Malhi and Singhi, 2001 study) | Identify current screening instruments for DD and ASD, create screening profiles, and provide recommendation for screening in LMIC. | A sample of more than 300 participants for each instrument | 0–7 years | Search Strategy of the tools (2014–2017); Inclusion and exclusion criteria and specific criteria for screening instruments. | The review identified 10 screening tools suitable to screen children in LMIC for ASD and 7 screening tools for DD. | PEDS is adapted and able to detect DD in LMIC. |
| Woolfenden et al., 2014 [59] | Peer reviewed | Systematic review | Chandigarh, India (Reference to Malhi and Singhi, 2001 study) | To understand the worldwide prevalence of parental concerns measured by PEDS that indicated developmental risks and associated risk factors. | 20 to 54,602 | Less than 1 month to 7 years and 11 months | Search Strategy of PEDS; Inclusion and exclusion criteria for study participants and review process. | 14% of parents raised concerns associated with a high risk of developmental problems, and 19% raised concerns about a moderate risk for developmental problems. | Eight studies of PEDS were conducted in low- and medium-income countries (including India). |
| Malhi and Singhi, 2001 [42] | Peer reviewed | Diagnostic test accuracy | Chandigarh, India | To identify the range of concerns that parents have about their child's development and its relationship to the child's developmental status. | 55 parent–child dyads | 6 to 60 months | Patients recruited through outpatient paediatric care in a tertiary care teaching hospital. | 38% of parents indicated no concerns, and 20% raised non-significant developmental concerns about their child's development. From these children, 90.6% passed the development screening. | Of the parents who expressed one or more significant developmental concerns about their child, 47.8% of these children failed the screening. In addition, 43% of the parents whose children failed developmental screening expressed medical concerns, 35.7% reported expressive language concerns, and 28% indicated global/cognitive concerns. |
| Malhi and Singhi, 2002 [43] | Peer reviewed | Diagnostic test accuracy | Chandigarh, India | To identify the range of concerns parents have about their child's development and evaluate the relationship between parent concern and the child's developmental status. | 79 parent–child dyads | 24 to 60 months | Patients recruited through outpatient paediatric care in a tertiary care teaching hospital; Two questionnaires used: PEDS and Developmental Profile II. | Parents' concerns about the developmental milestones of their child were moderately sensitive predictors of DD in children between 2 and 5 years. | The authors advised against using the PEDS as a substitute for standardized developmental screening measures because its specificity and sensitivity were lower than those reported by the US. The PEDS can be used as a pre-screening tool to find children who might need comprehensive developmental screening in outpatient settings. |

**Table 1.** *Cont.*

| | | | | | Population | | | | |
|---|---|---|---|---|---|---|---|---|---|
| Author | Article Type | Design | City in India | Aim/Purpose | Size | Age Range | Setting and Measures Used | Main Results | Key Findings |
| Glascoe, Malhi, and Singhi, 2001 and 2003 [42,51] | Grey literature | Letter to the authors | N.A. | | 79 parent–child dyads | 24 to 60 months | Patients recruited through outpatient paediatric care in a tertiary care teaching hospital; Two questionnaires used: PEDS and Developmental Profile II. | Parents' concerns about the developmental milestones of their child were moderately sensitive predictors of DD in children between 2 and 5 years. | The author's letter noted that the scoring method utilized in the paper was not clear. Given that DP-II has a propensity to overidentify developmental issues, the concurrent test utilized to evaluate the accuracy of PEDS is questionable. PEDS would gain from the use of a different scoring system. |
| Poon, Larosa, and Pai, 2010 [58] | Peer reviewed | Text and opinion study | | The paper discusses the prevalence of DD in children and recent literature regarding the benefits of early identification and benefits of developmental screening and surveillance. | Not specified | | N.A. | The review's key conclusions stated that it is important to pay attention to parents' worries while maintaining regular surveillance, integrating routine screening, making early referrals to paediatricians and therapists, and offering early intervention services and therapies that have been proven to be successful regardless of the medical diagnosis. | PEDS has been used in India. The tool has been translated to over 10 different languages and is completed by parents. |
| Mukherjee et al., 2014 [10] | Peer reviewed | Text and opinion Study | India | The aim of the article was to review existing tools for children under the age of five that were validated in India and to provide a purposed paradigm for developmental screening in office practice. | Not specified | Under the age of 5 years | N.A. | Tools developed in India lack psychometric properties and were developed by healthcare workers, and the screening tools developed in the US are costly and not easily accessible. | PEDS has been found reliable in developing countries. However, there is limited research from India. |

Note: PEDS was administered in English for all studies.

## 8. SDQ

From the 184 abstracts and titles screened for the SDQ scoping review, only eight full-text articles met the inclusion criteria (seven empirical studies and one literature review). The participants were recruited from different parts of India, and the children were up to 12 years old. The studies concluded that the SDQ was able to differentiate between different groups of children on the basis of their total difficulties score. No studies were conducted from 1990 to 2000, one (13%) study was conducted in both 2001–2005 and 2006–2010, two (25%) studies were conducted in 2011–2015, and four (50%) studies were conducted in 2016–2020. There was one (13%) study each in the format of a case-control study and a literature review, and there were three (38%) studies each using cross-sectional and longitudinal designs.

Of the three cross-sectional studies conducted in India using the SDQ, one study used a community sample [60], and two studies used clinical samples [61,62]. Bele et al. [60] assessed the prevalence of emotional and behavioural difficulties among children living in urban slums in Andhra Pradesh, India. Their study evaluated 370 children aged 5–10 years using the SDQ completed by parents. They concluded that for the children, residing in urban slums was significantly associated with behaviour problems. Boys had a higher risk of mental health difficulties than girls. Factors such as low nutrition, low socioeconomic status, financial constraints, and conflicts in the family were predictors of behaviour problems among the children.

Two cross-sectional studies examined the psychological health of children with acute lymphoblastic leukaemia and congenital heart disease [61,62]. Chari and Hirisave [61] assessed 40 children (20 children with acute lymphoblastic leukaemia and 20 children from a healthy group) aged 4 to 8 years in terms of psychiatric disturbance using the SDQ completed by parents. The study was carried out at the paediatric oncology ward at Kidwai memorial institute of Oncology, Bangalore, Karnataka. The results showed that children with acute lymphoblastic leukaemia demonstrated more disruptive behaviour and peer problems than healthy children.

Kiron [62] examined 242 children aged 10 years and below for the psychosocial impact of congenital heart disease. They used the Malayalam version of the SDQ, which was completed by parents at the Sree Chitra Tirunsal Institute of Medical Science and Technology, Kerala. They reported that children aware of their congenital heart disease had a higher total difficulty score on the SDQ than children unaware of their congenital heart disease. The children with awareness of their congenital heart disease also exhibited more behaviour problems and less prosocial behaviour than the children without awareness of their congenital heart disease [62].

The three longitudinal studies were carried out across several LMIC, including India. One was carried out in India alone [63], one in India and Vietnam [64], and one in Cambodia, Kenya, Tanzania, Ethiopia, and India [65]. The scoping review included these multi-country studies, as they met the inclusion criteria of including participants from India.

Malhotra et al.'s [63] study in Chandigarh, India, aimed to establish the incidence of psychological difficulties in 727 school children aged 4–11 years. At the six-year follow up, children with psychological disorders were compared to children with no psychological disorders on socio-demographic factors and psychological variables. The findings, based on parent reports, found no significant differences between the two groups on age, gender, or psychological parameters such as temperament, parental handling, life events, and IQ.

Trinh's [64] study compared the mental health impact of child labour in India and Vietnam. The study was of 1934 children aged 7–9 years assessed over a period of 15 years on parent-completed SDQ. The study concluded that the effect of child labour on the five dimensions of the SDQ was not uniform across the two countries. In Vietnam, children who participated in the labour market were likely to have worse conduct problems, hyperactivity, peer problems, and reduced prosocial behaviour compared to those who did not work. In India, child labour and mental health symptoms were significantly correlated with hyperactivity and reduced prosocial behaviour.

Huynh et al.'s [65] longitudinal study compared the psychological wellbeing of 2837 orphans and separated children in the age range of 6–12 years over 36 months in five LMIC (Cambodia, India, Kenya, Tanzania, and Ethiopia). The study used the self-report version of the SDQ and translated the questionnaire to its native languages in the five countries. The findings revealed no meaningful difference in the SDQ total difficulties score across care settings (residential versus community-based) or between orphaned and separated children in residential care settings. In this study, orphans were defined as children who had lost one or both parents, and separated children were defined as children separated from their parents with no expectation of return. This study demonstrated that in under-resourced societies in LMIC, orphaned and separated children's overall wellbeing may depend on the quality of care rather than the type of care setting itself [65].

A prospective case-control study was conducted in Mumbai, Maharashtra, to study the impact of neurobehavioral disorders in children with and without epilepsy [66]. The children, aged 5–12 years, were classified on the etiologic classification: epilepsy, epilepsy control, irregular school attendance, and school dropouts. The 222 children with epilepsy were matched with 226 non-epileptic children on age, gender, and socioeconomic status. The parent version of the SDQ was administered in the Indian language to screen for neurobehavioral disorders. The authors reported that 63% of the children with epilepsy had emotional problems and abnormal conduct scores, high hyperactivity, poor peer relations, and poor pro-sociality, leading to low school attendance. The SDQ total difficulties score was abnormal in 39%, borderline in 16%, and normal in 45% of the cases for children with epilepsy. It was abnormal for 8%, borderline for 3%, and normal for 89% of the cases for children without epilepsy.

A literature review (grey literature) by Galab et al. [67] focused on childhood poverty in Andhra Pradesh, India, and how the national policies have impacted the state. The report used the parent and translated version (Telegu) of the SDQ to assess the mental health problems of children aged 1–8 years. The report concluded that nearly 20% of children were classified as abnormal and 20% as borderline However, the author cautioned that the SDQ had not been validated for use in Andhra Pradesh and that local normative data were not available [55]. Table 2 summarises the studies conducted using SDQ in India. The table reports the authors, the article type, the design of the study, the city/state where the study was conducted, the aim of the study, the population setting (size and age range), the measures used, the results, and the key findings of the study.

**Table 2.** Summary of studies on the Strength and Difficulties Questionnaire (SDQ) use in India.

| | | | | | Population | | | | |
|---|---|---|---|---|---|---|---|---|---|
| Author | Article Type | Design | City in India | Aim/Purpose | Size | Age Range | Setting and Measures Used | Outcome | Key Findings |
| Bele et al., 2013 [60] * | Peer reviewed | Cross-sectional study | Gauthaminagar in Karimnagar district of Andhra Pradesh | To estimate the prevalence of emotional and behavioural disorders using standardised instruments among children in urban slums. | N = 370 | 5–10 years | Emotional and behavioural problems among children were evaluated using the Strength and Difficulties Questionnaire (SDQ), and depression was assessed using Patient Health Questionnaire (PHQ-9). | On at least one SDQ domain, 22% of the children scored abnormally. The children's behavioural issues and poorer academic achievement were found to be significantly correlated. | SDQ scores and mean values for affected and unaffected groups were compared, and a significant variance was found in the total problems score in the affected group (borderline and abnormal score) compared to the unaffected group. |
| Malhotra et al., 2009 [63] ** | Peer reviewed | Prevalence study/Longitudinal study | Chandigarh, India | To establish the incidence of psychiatric disorders in school children in India. | N = 873 | 4–11 years | Rutter B (teachers rating), Childhood Psychopathology Measurement Schedule, Temperament Measurement Scale, Parent Handling Questionnaire, Parent Interview Schedule, Life event scale for Indian children, and SDQ. | A total of 20 of the 186 children that were monitored had a psychological condition. In terms of age, gender, and psychological (temperament, parental handling, life stress, and IQ) factors at baseline, children with the disorder at follow-up did not vary from those without it. | Children scoring above a cut-off score on the SDQ ($\geq 14$) were clinically examined by a psychiatrist at home or at the clinic. |
| Trinh, 2020 [64] * | Peer reviewed | Longitudinal study | India and Vietnam | To study the mental health impact of child labour. | N = 978 children in Vietnam and 956 children in India | 7–9 years | SDQ to measure child mental health and child participation in the labour market was assessed by understanding if the child has undertaken any activity to earn money. | Child labour did not uniformly affect the five dimensions of the SDQ. Compared to children who did not work, those who participated in the labour market in Vietnam were more likely to experience conduct problems, hyperactivity, peer issues, and less prosocial behaviour. The outcomes for working children in Vietnam were noticeably lower regarding peer issues and less prosocial behaviour. Hyperactivity and a decline in prosocial behaviour were significantly linked with child labour and mental health symptoms in India. | In the five scales of the SDQ, peer problems and prosocial behaviour were found to be significantly impacted by working in both countries. |

**Table 2.** *Cont.*

| | | | | | Population | | | | |
|---|---|---|---|---|---|---|---|---|---|
| Author | Article Type | Design | City in India | Aim/Purpose | Size | Age Range | Setting and Measures Used | Outcome | Key Findings |
| Anita et al., 2016 [66] ** | Peer reviewed | Prospective case control study | Mumbai, India | To study the prevalence, type, and impact of neurobehavioral disorders in children with and without epilepsy. | N = 222 | 5–12 years | SDQ was assessed in four groups: epilepsy, epilepsy control, irregular school attendance, and school dropout. | The study revealed that 14.4% of children with epilepsy during schooling had learning problems, and 10.3% had behavioural problems compared to non-epileptics. In addition, 63% of the people with epilepsy had emotional difficulties and abnormal conduct scores. High hyperactivity, poor peer relations, and poor pro-social behaviour led to low school attendance in 35% of epileptic patients. | Screening of cases and controls with the SDQ-P (parent version) was conducted, and the total difficulties score was abnormal in 39% of cases and 7.9% of controls, and it was normal in 44.5% of cases and 88.9% of controls. |
| Chari and Hirisave, 2020 [61] * | Peer reviewed | Cross-sectional study | Bangalore, India | To examine the psychological health of young children undergoing treatment for acute lymphoblastic leukaemia. | N = 40 | 4–8 years | SDQ to assess psychiatric disturbances, feeling cards to examine the subject's current emotional state, and teddy bear's picnic to examine personal construct. | Children with ALL (acute lymphoblastic leukaemia) were reported on SDQ to display more behavioural disturbances. | On the SDQ, there were significant differences between groups in total difficulties, conduct, and peer problems. However, median scores were in the normative range. Children with ALL demonstrated more disruptive behaviours and peer problems than healthy children. |
| Galab et al., 2003 [67] *** | Grey literature | Literature review | Andhra Pradesh | The report provides a brief literature on childhood poverty in Andhra Pradesh in India and explains how national policies have impacted childhood poverty in that state. | N = 3000 | 1–8 years | The SDQ was used to assess the mental health of children of 8 years of age. | The Young Lives results reported that nearly 20% of children were classified as abnormal and 20% as borderline. However, the authors recommended that these results should be interpreted with caution since the SDQ has not been validated in Andhra Pradesh and normative data is not available. | Previously, the SDQ had not been validated in Andhra Pradesh, and normative data were unavailable. The study was the first to use the SDQ in Andhra Pradesh, and the findings suggest that child mental health issues may be a potential problem, especially in the rural areas of Andhra Pradesh, where the prevalence of abnormal cases was over 20%. |

**Table 2.** *Cont.*

| | | | | | Population | | | | |
|---|---|---|---|---|---|---|---|---|---|
| Author | Article Type | Design | City in India | Aim/Purpose | Size | Age Range | Setting and Measures Used | Outcome | Key Findings |
| Huynh et al., 2019 [65] ** | Peer reviewed | Longitudinal study | Five low- and middle-income countries: Battambang District, Cambodia; Nagaland and Hyderabad, India; Bungoma District, Kenya; Kilimanjaro Region, Tanzania; and Addis Ababa, Ethiopia. Children were enrolled between 2006 and 2008 and followed biannually | To examine the psychological wellbeing of orphans and separate children in under-resourced societies in LMIC and to associate quality of care with children's psychosocial wellbeing. | N = 2013 (923 residential care and 1090 community-based sample) | 6–12 years at baseline; 36 months of follow-up data | Quality of care was assessed using the Child Status Index, and child's psychosocial wellbeing was assessed using the SDQ. | Child psychosocial well-being across different levels of quality of care showed negligible differences between residential- and community-based care settings, suggesting the critical factor in child well-being is quality of care rather than environment of care. | When the authors controlled the orphan's gender, status, and age, components of quality of care predicted SDQ total difficulties better than care setting. Mean SDQ total difficulties scores across "high" and "low" quality of care showed differences between care settings to be minimal. There were no meaningful differences in SDQ total difficulties across care settings. Orphans and separated children (OCS) in residential care settings had higher SDQ total difficulties scores than in community-based settings. |
| Kiron, 2012 [62] *** | Grey literature | Cross-sectional study (Project 2) | Sree Chitra Tirunal Institute of Medical Science and Technology, Kerala, India | To analyse whether those children who grow up without being aware of their congenital heart disease have any psychosocial advantage over those children who grow up being aware that they have undergone a major interventional procedure for their congenital heart disease. | N = 242 (only 147 parents responded to SDQ) | Less than 10 years | The SDQ was used to ascertain the impact of CHD in children. | On being assessed with the total difficulties score, children who were aware of their congenital heart disease were at substantial risk of clinically significant problems compared to the other group. | Children not aware of their CHD had significantly lower levels of problems compared to children who were aware and had experienced CHD. In addition, children in the first group were higher in prosocial behaviour compared to the second group. |

Note: * demonstrates SDQ administered in English, ** demonstrates SDQ administered in Hindi and *** demonstrates SDQ administered in the regional languages.

## 9. Summary

The key findings addressed the three aims of the scoping review. First, there is scant published literature on the use of the PEDS in India. Most of the literature exists in the form of text- and opinion-based evidence that emphasised the lack of screening in India and the limited use of the PEDS as a tool to screen children for DD. Furthermore, only one study briefly mentioned the use of translated versions of PEDS questionnaires [37]. Overall, the articles highlighted the importance of early identification of children with DD and listening to parents' concerns through regular surveillance and screening. The SDQ compared mental health across different LMIC and was administered more frequently than PEDS to screen children in India.

Second, the characteristics of the participants included in the PEDS study were children aged 6–60 months with typical development living in Chandigarh, India. The SDQ studies comprised children aged 0–12 years from clinical and community samples recruited from different parts of the country.

Third, empirical studies using PEDS reported the tool as having below acceptable sensitivity and specificity. The SDQ studies reported that the tool was effective in differentiating groups of individuals based on SEL and behavioural concerns. Since India is a diverse country with many regional languages, the studies that used the SDQ catered to different population types and translated the questionnaire to regional and national languages.

### 9.1. Limitations of the Existing Research

The limited number of studies identified in this scoping review were conducted in different parts of India and examined the use of the PEDS and SDQ with children. Empirical research on the use of the PEDS in India is scarce (Table 1). The PEDS:DM was not used together with the PEDS in any of the studies. The studies that reported using translated versions of the PEDS in India provided limited to no information on the psychometric properties of the translated versions. Marlow et al. [37] noted that Malhi and Singhi's [42,43] study translated PEDS to Hindi. However, the original article does not provide this information.

The SDQ has been used more frequently than the PEDS in India (Table 2). However, most of the studies did not consider teacher evaluation and lacked transparency in the translation process of the SDQ. Two studies (grey literature) translated the SDQ to the relevant regional language [62,67]. Two studies indicated that they had translated the SDQ to the Native or Indian language without explicitly saying which language [65,66], and three studies did not translate the SDQ to Hindi [60,61,64]. Only one study explicitly indicated that they had used the Hindi version of the SDQ in their research [63]. The psychometric properties of the translated questionnaire in the regional and national languages of India must be considered with caution. The recommended cut-off value for the UK cannot be assumed to be suitable for use in countries where cultural differences exist [37].

### 9.2. Strengths, Limitations, and Implications of this Scoping Review

Strengths of this review are its comprehensive search strategy and its addressing of a broad research question related to the use of two screening tools in the population of India. The review process evaluated the quality of the studies that met the inclusion criteria for the PEDS and the SDQ. However, the review only assessed papers in English that were published within a certain year range. Future studies could possibly include studies in other languages used in India.

## 10. Conclusions

In terms of screening children for DD and SEL concerns, India lacks both a literature base and evidence of practice. Evidence on the use of the PEDS and SDQ suggests that these screening tools have not been widely used with Indian children. Therefore, the translation and administration of the PEDS, PEDS:DM, and SDQ will ensure that these screening tools are relevant and applicable to the Indian population. Furthermore, concurrent use of these

tools will provide a better understanding of the relationship of DD with SEL concerns among children. It is important that research is undertaken and published to address the current gap in local literature and practice.

## 11. Implications for Practice

Due to the limited availability of suitable measures, screening children for developmental delay and social–emotional learning is a challenge in India. A scoping review was conducted on the use of the Parent's Evaluation of Developmental Status (PEDS) and the Strength and Difficulties Questionnaire (SDQ) with children in India. There is an absence of research demonstrating the complementary use of both measures to provide a holistic screening of children.

**Author Contributions:** H.S. wrote the manuscript under the supervision of L.S. and N.V.M. H.S. and N.V.M. conducted the data analysis. All authors contributed to and reviewed the manuscript. All authors have read and agreed to the published version of the manuscript.

**Funding:** This research received no external funding.

**Institutional Review Board Statement:** Not applicable.

**Informed Consent Statement:** Not applicable.

**Data Availability Statement:** Not applicable.

**Conflicts of Interest:** The authors declare no conflict of interest.

**Research Ethics:** This is a literature review; research ethics approval was not required.

## Appendix A

**Table A1.** Data Extraction Table.

| Background Information |
|:---:|
| Author and Date of Publication |
| Article Type |
| Type |
| City/State |
| Aim of the Study |
| Sample |
| Number of Participants |
| Age Range |
| Setting and Measures used |
| Results |
| Main Results |
| Key Findings |

**Appendix B**

**Table A2.** Preferred Reporting Items for Systematic reviews and Meta-Analyses extension for Scoping Reviews (PRISMA-ScR) Checklist.

| SECTION | ITEM | PRISMA-ScR CHECKLIST ITEM | REPORTED ON PAGE # |
|---|---|---|---|
| **TITLE** | | | |
| Title | 1 | Identify the report as a scoping review. | Page 1 |
| **ABSTRACT** | | | |
| Structured summary | 2 | Provide a structured summary that includes (as applicable): background, objectives, eligibility criteria, sources of evidence, charting methods, results, and conclusions that relate to the review questions and objectives. | Page 2 |
| **INTRODUCTION** | | | |
| Rationale | 3 | Describe the rationale for the review in the context of what is already known. Explain why the review questions/objectives lend themselves to a scoping review approach. | Page 5 |
| Objectives | 4 | Provide an explicit statement of the questions and objectives being addressed with reference to their key elements (e.g., population or participants, concepts, and context) or other relevant key elements used to conceptualize the review questions and/or objectives. | Page 7 |
| **METHODS** | | | |
| Protocol and registration | 5 | Indicate whether a review protocol exists; state if and where it can be accessed (e.g., a Web address); and if available, provide registration information, including the registration number. | OSF |
| Eligibility criteria | 6 | Specify characteristics of the sources of evidence used as eligibility criteria (e.g., years considered, language, and publication status), and provide a rationale. | Page 8 |
| Information sources * | 7 | Describe all information sources in the search (e.g., databases with dates of coverage and contact with authors to identify additional sources) as well as the date the most recent search was executed. | Page 8 |
| Search | 8 | Present the full electronic search strategy for at least one database, including any limits used, such that it could be repeated. | Page 8–10 |
| Selection of sources of evidence † | 9 | State the process for selecting sources of evidence (i.e., screening and eligibility) included in the scoping review. | Page 8–10 |
| Data charting process ‡ | 10 | Describe the methods of charting data from the included sources of evidence (e.g., calibrated forms or forms that have been tested by the team before their use, and whether data charting was done independently or in duplicate) and any processes for obtaining and confirming data from investigators. | Page 10–13 |
| Data items | 11 | List and define all variables for which data were sought and any assumptions and simplifications made. | Page 13 |
| Critical appraisal of individual sources of evidence § | 12 | If done, provide a rationale for conducting a critical appraisal of included sources of evidence; describe the methods used and how this information was used in any data synthesis (if appropriate). | - |
| Synthesis of results | 13 | Describe the methods of handling and summarizing the data that were charted. | Page 11–12 |
| **RESULTS** | | | |
| Selection of sources of evidence | 14 | Give numbers of sources of evidence screened, assessed for eligibility, and included in the review, with reasons for exclusions at each stage, ideally using a flow diagram. | Page 16–18 and 23–25 |
| Characteristics of sources of evidence | 15 | For each source of evidence, present characteristics for which data were charted, and provide the citations. | Page 16–18 and 23–25 |
| Critical appraisal within sources of evidence | 16 | If done, present data on critical appraisal of included sources of evidence (see item 12). | - |

**Table A2.** *Cont.*

| SECTION | ITEM | PRISMA-ScR CHECKLIST ITEM | REPORTED ON PAGE # |
|---|---|---|---|
| Results of individual sources of evidence | 17 | For each included source of evidence, present the relevant data that were charted that relate to the review questions and objectives. | Page 16–18 and 23–25 |
| Synthesis of results | 18 | Summarize and/or present the charting results as they relate to the review questions and objectives. | Page 26 |
| **DISCUSSION** | | | |
| Summary of evidence | 19 | Summarize the main results (including an overview of concepts, themes, and types of evidence available), link to the review questions and objectives, and consider the relevance to key groups. | Page 13–15 and 19–22 |
| Limitations | 20 | Discuss the limitations of the scoping review process. | Page 26–27 |
| Conclusions | 21 | Provide a general interpretation of the results with respect to the review questions and objectives, as well as potential implications and/or next steps. | Page 27–28 |
| **FUNDING** | | | |
| Funding | 22 | Describe sources of funding for the included sources of evidence as well as sources of funding for the scoping review. Describe the role of the funders of the scoping review. | N.A. |

JBI = Joanna Briggs Institute; PRISMA-ScR = Preferred Reporting Items for Systematic reviews and Meta-Analyses extension for Scoping Reviews. * Where *sources of evidence* (see second footnote) are compiled from, such as bibliographic databases, social media platforms, and Web sites. † A more inclusive/heterogeneous term used to account for the different types of evidence or data sources (e.g., quantitative and/or qualitative research, expert opinion, and policy documents) that may be eligible in a scoping review as opposed to only studies. This is not to be confused with *information sources* (see first footnote). ‡ The frameworks by Arksey and O'Malley (6) and Levac and colleagues (7) and the JBI guidance (4, 5) refer to the process of data extraction in a scoping review as data charting. § The process of systematically examining research evidence to assess its validity, results, and relevance before using it to inform a decision. This term is used for items 12 and 19 instead of "risk of bias" (which is more applicable to systematic reviews of interventions) to include and acknowledge the various sources of evidence that may be used in a scoping review (e.g., quantitative and/or qualitative research, expert opinion, and policy document). *From:* Tricco, A.C.; Lillie, E.; Zarin, W.; O'Brien, K.K.; Colquhoun, H.; Levac, D.; Moher, D.; Peters, M.D.J.; Horsley, T.; Weeks, L.; et al. PRISMA Extension for Scoping Reviews (PRISMAScR): Checklist and Explanation. *Ann. Intern. Med.* **2018**, *169*, 467–473. https://doi.org/10.7326/M18-0850. [68].

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
