# Peer review of "Parents’ Evaluation of Developmental Status and Strength and Difficulties Questionnaire as Screening Measures for Children in India: A Scoping Review"

_pediatrrep, doi:10.3390/pediatric15010014_

Round 1

Reviewer 1 Report

1. This is important work as part of a need for research focussed on and undertaken in India as a LMIC in relation to identifying children with developmental disabilities. Your method is clearly described and has followed PRISMA and JBI guidance. It is important to highlight the need for more evidence/ research in this field.

2. Some of the references in the introduction are too old (e.g. [1] and [6], I am not sure if you need [1] - If DSM and ICD are used in India, these would be the better definitions to use.

3. Ref [6] language and descriptions is outdated. Although the Neurodiversity movement and ND paradigm may be less visible in LMIC, there is a strong international message of the need to understand neurodevelopmental differences rather than impairments or disorders and that difficulties arising for people with neurodevelopmental conditions arise because of the interaction between the individual and their physical and social environment.

3. Please reword phrase 'cannot read non verbal and ...' to reflect the difference not deficit concept within a Neurodiversity paradigm. For example, individuals may have difficulties with social relationships when compared to neurotypical peers and have differences in their reading of neurotypical nonverbal and subtle social cues. 

4. I recommend adding some more context about the population of children and families in India - what proportion have access to screening and child health surveillance? Is it universally available in any states and what happens when developmental disabilities are identified. Is there universal education and how are educators involved in screening and surveillance... ie if there was a an agreed and recommended set of low or no cost tools how would they be deployed. What has changed in the last 5-10 years?

5. You chose a long time frame (from 1990) and initially I was concerned about the relevance of data from so long ago in a country that has changed so much in that time. However I see that in fact there was little research before 2000. I would be interested in a comment in the paper about how the societal/ health/ education context has changed in the last 2 decades and  reference to the need for up to date research.

6. You explain well that the cost of many tools developed in the West prevents them being a valid solution for India. However please can you give a comment about any quality/ content differences between the PEDS and SDQ and the others. Are they similar in length, do they cover the same topics, what are the similarities and differences? 

7. In the introduction section, you describe the sensitivity and specificity of the PEDS and SDQ - please add statement/ comment/ interpretation of this data.

8. Aims - looking at the second aim, can you 'ensure' that tools used are less costly? How would you do this with a paper? Are you taking other steps not reported? Perhaps your aim is more to 'promote the use of valid, reliable and accessible low cost tools'?

9. Line 187 - missing a full stop

10. Table 1 - in my reading view, the final column and larger font than the rest of the table, reducing readability

11. In terms of next steps/ outcomes/ recommendations - should you consider the need to develop new and up to date surveillance and screening tools for the context of India, funded through a source which allows for free access and adaptations for different contexts of India (child health surveillance, teacher/ school identification, online resources for parents with such access, social care settings/orphanages)

12. what are the implications of earlier and wider identification of need? what information and support or care should be matched to identification?

Reviewer 2 Report

Sheel et al. have done a narrative review article on the Parents’ Evaluation of Developmental Status (PEDS), and Strength and Difficulties Questionnaire(SDQ) as Screening Measures for Children in India. This will add one more article to the current literature. Overall, the article is written well in a well-elaborated manner with relevant references. 

Some of the specific comments are:

The abstract is written well, and it reflects the facts in the article.

The introduction is well-written, and relevant references are cited. We have a few suggestions.

Page 1, line 22: disorders of the developing nervous system [1] (p. 4).

Author can consider removing (p.4). The citation should look similar throughout the article.

Page 1, line 43: Developmental screening "is a brief assessment procedure designed….”

Author can consider rephrasing it as "Developmental screening is a brief assessment procedure designed…”

Page 1, line 44: assessment" [9] (p. 527). 

Author can consider removing (p.527). The citation should look similar throughout the article.

The introduction seems longer. The author can consider splitting the introduction content under a few subheadings. 

Page 2 line 93: The interrater reliability was .95 and the…

Consider rewriting all the numbers with decimal with 0 before decimal. Eg. 0.95 instead of .95

Search Strategy:

Page 4 , line 182: the PEDS: DM was not mentioned in any articles in the initial screening.

Can the author confirm this? I found a few published articles about PEDS:DM. Eg. Mukherjee, Sharmila Banerjee et al. “Diagnostic Accuracy of Parents' Evaluation of Developmental Status (PEDS), PEDS Developmental Milestones, and PEDS Combined in Indian Children Aged Less than 2 Years.” Indian journal of pediatrics vol. 89,5 (2022): 459-465. doi:10.1007/s12098-020-03651-y 

Please fix grammar errors and spelling typos in the article. 

It would have been better if the author could have included a few words about PEDS:DM and other screening tests such as Baroda Development Screening Test (BDST) and Developmental Assessment Scale for Indian Infants (DASII).
